# Performance of the ATMOS41 All-in-One Weather Station for Weather Monitoring

**DOI:** 10.3390/s21030741

**Published:** 2021-01-22

**Authors:** Olga Dombrowski, Harrie-Jan Hendricks Franssen, Cosimo Brogi, Heye Reemt Bogena

**Affiliations:** Agrosphere (IBG-3), Forschungszentrum Jülich GmbH, 52425 Jülich, Germany; o.dombrowski@fz-juelich.de (O.D.); h.hendricks-franssen@fz-juelich.de (H.-J.H.F.); c.brogi@fz-juelich.de (C.B.)

**Keywords:** ATMOS41, all-in-one, weather monitoring, low-cost, weather station comparison

## Abstract

Affordable and accurate weather monitoring systems are essential in low-income and developing countries and, more recently, are needed in small-scale research such as precision agriculture and urban climate studies. A variety of low-cost solutions are available on the market, but the use of non-standard technologies raises concerns for data quality. Research-grade all-in-one weather stations could present a reliable, cost effective solution while being robust and easy to use. This study evaluates the performance of the commercially available ATMOS41 all-in-one weather station. Three stations were deployed next to a high-performance reference station over a three-month period. The ATMOS41 stations showed good performance compared to the reference, and close agreement among the three stations for most standard weather variables. However, measured atmospheric pressure showed uncertainties >0.6 hPa and solar radiation was underestimated by 3%, which could be corrected with a locally obtained linear regression function. Furthermore, precipitation measurements showed considerable variability, with observed differences of ±7.5% compared to the reference gauge, which suggests relatively high susceptibility to wind-induced errors. Overall, the station is well suited for private user applications such as farming, while the use in research should consider the limitations of the station, especially regarding precise precipitation measurements.

## 1. Introduction

Weather monitoring plays a central role in the understanding of the hydrological cycle, weather forecasting, risk assessment and management as well as agricultural planning, the administration of natural resources, climate change studies and other public and private interests. Despite the fact that modern automatic weather station networks are typically well developed in high-income countries, data quality and station coverage are often limited in low-income countries due to high instrumentation and maintenance costs [1,2,3]. Consequently, resources and trained personnel to set up and maintain a sufficient number of stations are lacking to adequately cover the spatiotemporal variability of meteorological variables [4,5]. Additionally, growing interest in microclimate monitoring for precision agriculture [6,7,8] or urban climate and heat island studies [9,10] requires weather stations that are inexpensive, efficient, and provide local and reliable data for modelling applications. Ideally, the design of such weather stations meets the following criteria: (i) robustness to reduce calibration frequency; (ii) compact design for ease of handling and to minimize sensor damage; (iii) low maintenance; (iv) low power requirements; (v) low cost; (vi) compatibility with different logger systems; (vii) wireless communication.

With the increasing use of wireless sensor networks [11], various non-standard low-cost weather monitoring systems have been developed in the past few years using a wide range of sensor hardware and different microcontroller architectures, such as Arduino [12,13,14] or Raspberry Pi [7,15,16]. These stations can be very cost effective, with prices of several hundred Euros [3], but they often lack adequate calibration and testing, raising concerns about the accuracy, precision, and reliability of the collected data [17]. However, information on data quality in terms of both the accuracy and repeatability of such low-cost weather stations is crucial for modelling applications and decision-making [18,19]. Furthermore, designing these non-standard devices requires sufficient knowledge of the associated hardware and software for installation and maintenance, while the moving parts may be susceptible to failure. Commercial all-in-one weather stations, e.g., [20,21,22], that incorporate multiple sensors in a single unit can be a viable alternative as they are easily deployable, generally cheaper than standard weather stations composed of individual sensors, and include manufacturer reported accuracy, precision, and calibration details. On the downside, they are less flexible in terms of adding or exchanging sensors and may suffer from interference between sensors due to their compact architecture [23]. Nonetheless, their plug-and-play principle and the compact design are clear advantages since they facilitate non-expert use and make them suitable for continuous deployment in rural or remote areas.

This study focuses on the assessment of the ATMOS41 all-in-one weather station that holds 12 embedded sensors, developed and produced by METER Group, Inc. The station is currently used in sub-Saharan Africa to improve crop production of maize [24] and to build the Trans-African Hydro-Meteorological Observatory (TAHMO) network [25]. TAHMO aims at installing 20,000 hydro-meteorological stations across sub-Saharan Africa and collected data will be used for educational purposes at local schools as well as aid in scientific modelling, early warning systems, and the analysis of water availability [26]. Furthermore, the ATMOS41 has recently found applications in crop research and private sector sensor networks of various industrialized countries. In Portugal, the station is being used in the development of a forest monitoring system for fire detection [27] and in the field of smart agriculture to improve vineyard management practices [28]. In addition, the ATMOS41 is part of the Montana Mesonet monitoring stations in the Upper Missouri River Basin, where collected data are used for drought detection and natural resource management, amongst others [29]. Further applications include the investigation of crop water stress in apple orchards at Washington State University [30] and the estimation of the plant growth status of paddy rice in Japan [31].

METER and partners provide reports of calibration and sensor performance tests for the complete weather station or for individual components performed in the lab or in outdoor testbeds of the METER Pullman campus [32]. Furthermore, [33] conducted a first-order performance analysis of the early version of the station. The study compared 6 months of data recorded in 2017/2018 by the ATMOS41 station against a weather station of the Institute of Atmospheric and Climate Science (IAC) of ETH Zurich and a SwissMetNet solar radiation station located at 2.5 km distance from the test site. Overall, the ATMOS41 showed similar performance to the IAC station, but the authors suggested that further tests are needed.

Since its first release in 2017, several improvements of the ATMOS41 station were developed, some of which directly affect the measurement of certain variables and the overall performance of the station [34]. These changes include: (i) improved sensor geometry to avoid adverse effects on wind measurements caused by heavy rain, (ii) improved sensor firmware and wind sensing algorithm, (iii) upgraded sensors and the addition of a secondary calibration for relative humidity and atmospheric pressure.

Considering the wide use of the ATMOS41 weather station for small- and large-scale weather monitoring in sub-Saharan Africa [24,25,26] as well as industrialized countries [27,28,29,30,31], independent testing under “out of the lab” conditions can provide further insight and eventually identify possible limitations of the ATMOS41 station. In this way, a thorough performance assessment can inform private costumers and research organizations regarding the potential fields of application and provide impulses for further hardware or software developments. Therefore, the aim of this study is to carry out such in-depth assessment through direct comparison to an independent, high-performance weather station as well as the inter-comparison of multiple ATMOS41 stations. Within this context, the following questions will be addressed:What is the quality of weather data from the ATMOS41 weather station?What systematic or random errors affect the ATMOS41 station?How well does the ATMOS41 station perform compared to a high precision, high quality weather station?What are the limitations of the ATMOS41 station?

## 2. Materials and Methods

### 2.1. ATMOS41 All-in-One Weather Station

The ATMOS41 is an all-in-one weather station developed by METER Group, Inc. (Pullman, WA, USA). The device is rather inexpensive for developed countries (below EUR 2000), has a compact design with no moving parts, and can be mounted with minimal effort to ensure easy deployment in a variety of terrains and locations. The station has 12 embedded sensors that measure standard weather variables, namely solar radiation, precipitation, air temperature, relative humidity, atmospheric pressure, wind speed and direction, plus additional parameters such as lightning strike count or compass heading. Further characteristics of the station are summarized in Table 1.

### 2.2. Reference Weather Station

The performance of the ATMOS41 weather station was evaluated through a comparison with measurements from a meteorological station that serves as a backup station for the official Selhausen (C1) measurement site [35], which is part of the Integrated Carbon Observation System (ICOS) [36]. The backup station, hereafter referred to as ICOS-bkp, consists of individual, high-quality sensors that fully comply with the ICOS standard. This standard specifies minimum requirements for sensor selection as recommended by the World Meteorological Organization (WMO) [37] and includes detailed descriptions for measurement and calibration processes as well as regular maintenance [38]. ICOS measurement uncertainty requirements are based on the “achievable uncertainty” that can be expected in operational practice, as specified in the WMO Guide No. 8 [37]. The total equipment costs for an ICOS level one station are estimated at EUR 10,000 [39], including the costs of logger and tripod (ca. EUR 1800 for the Selhausen station). The cost of weather sensors used at an ICOS station is hence more than four times the cost of an ATMOS41 device. The ICOS-bkp station records instantaneous values for solar radiation, temperature, and relative humidity at an interval of 20 s and an installation height of 2.5 m. Precipitation is recorded at a height of 1 m above ground, and a 10 min accumulated value recorded at a separate data logger was used for the comparison with the ATMOS41 stations.

Atmospheric pressure, wind speed, and wind direction are only recorded at the main ICOS station but are not recorded at the backup station. For the comparison of wind speed and direction, data recorded by a Vaisala WXT520 weather transmitter (Vaisala Corporation, Helsinki, Finland) were used. This instrument is installed at a height of 2 m above ground next to the ICOS-bkp station and records data for the SE_BDK_002 station of the Terrestrial Environmental Observatories network (TERENO) [40] at a 10 min interval. The Vaisala WXT520 meets the high accuracy and precision standards specified by ICOS for wind speed and direction but has a measurement uncertainty of ±0.5 hPa for atmospheric pressure instead of the ±0.3 hPa required by ICOS standards. Therefore, the atmospheric pressure sensor at the main ICOS was used as a reference to the ATMOS41 stations.

### 2.3. Experimental Setup

Data were collected from 23 April to 5 July 2020 (73 days) in Selhausen, Germany (50.87 N 6.45 E) at an altitude of 103 m.a.s.l. The area is characterized by a temperate maritime climate with a mean annual air temperature of 10 °C and annual precipitation of 700 mm. The site is located in an agricultural area with the dominant crops being sugar beet, winter wheat, and winter barley [36].

Three ATMOS41 weather stations (hereafter referred to as Atmos1, Atmos2 and Atmos3) were set up next to the Vaisala and ICOS-bkp stations. Atmos1 is the first generation of the station, purchased in 2017, and was previously deployed for a period of less than 6 months. Atmos2 and Atmos3 are the latest versions of the station, purchased in 2020, and used for the first time in this study. All three ATMOS41 stations were mounted in a row and installed at 2 m above ground (Figure 1). The stations were oriented north and levelled according to the user manual [41] to ensure accurate measurements of wind direction, precipitation, and solar radiation. Cumulative or instantaneous data were recorded at a 10 min interval for precipitation and all other variables, respectively. The ATMOS41 stations were connected to a CR1000X data logger (Campbell Scientific Ltd., Logan, UT, USA) which was powered via a 12 V battery connected to a battery charger.

Details on the sensors that measured each variable for the ATMOS41 and for the ICOS-bkp, ICOS or Vaisala stations, including approximate costs for individual sensors used at the reference stations, are listed in Table 2. The accuracy of most weather sensors used in the ATMOS41 station, as stated by the manufacturer, is compliant with the “achievable uncertainty” standard used by ICOS, with the exception of the air temperature and atmospheric pressure sensor (ICOS standard of ±0.1 °C and ±0.3 hPa, respectively).

### 2.4. Performance Analysis

Python software (version 3.7.6, Python Software Foundation) was used for the graphical and statistical evaluation of the data quality and performance of the ATMOS41 weather station. Data were checked for consistency and erroneous measurements were removed manually. Wind speed and relative humidity were computed according to the procedure described in the ATMOS41 user manual [41]. Data from the ICOS-bkp and ICOS station were resampled to 10 min instantaneous data for comparison to the ATMOS41 data. Measured atmospheric pressure was corrected for the difference of 3.7 m in observation height (combination of elevation and sensor installation height) between the instrument locations using the barometric formula, while the effect of the distance of 350 m between the stations was considered negligible. Graphical evaluation included time series plots and scatterplots for each parameter. Additionally, residual plots and correlation matrices were obtained and analysed. Residuals were calculated by subtracting the value obtained at the ATMOS41 stations from the value measured at the reference station using hourly mean values (hourly sums for precipitation).

The statistical analysis of solar radiation only considered daytime values as measured nighttime solar radiation was zero. For the evaluation of measured precipitation, all time steps without precipitation were discarded. Statistical analysis of precipitation additionally included an event-based approach using a minimum rainfall amount of ≥ 0.2 mm/event and a minimum inter-event time of 1 h.

For statistical comparison, the Arithmetic Mean (μ) of the measured variables was calculated. Other metrics included the Coefficient of Determination (R^2^, Equation (1)) as a measure of agreement between two stations. The Root Mean Square Error (RMSE, Equation (2)) was used as a measure of the difference between two stations. The RMSE is sensible to outliers since higher weights are given to larger deviations between two stations [42]. The Mean Bias Error (MBE, Equation (3)) was used as a measure of the average error between a station and the reference, with positive values indicating an overestimation and negative values indicating an underestimation. The MBE should be used in combination with other metrics as it is subject to cancellation errors since the sum of positive and negative values may result in a smaller MBE [43]. Lastly, the Mean Absolute Error (MAE, Equation (4)) was used as a measure of the absolute difference of a measurement compared to the reference measurement. It is not subject to cancellation errors and is less sensitive to outliers compared to the RMSE [42].
(1)$$R^{2}=1-\frac{\sum_{i=1}^{N}{\left(y_{i}-{\hat{y}}_{i}\right)}^{2}}{\sum_{i=1}^{N}{\left({\hat{y}}_{i}-\overset{-}{y}\right)}^{2}}$$
(2)$$RMSE=\sqrt{\frac{1}{N}\sum_{i=1}^{N}{\left(y_{i}-{\hat{y}}_{i}\right)}^{2}}$$
(3)$$MBE=\frac{\sum_{i=1}^{N}\left(y_{i}-{\hat{y}}_{i}\right)}{N}$$
(4)$$MAE=\frac{1}{N}\sum_{i=1}^{N}\left|y_{i}-{\hat{y}}_{i}\right|$$
where *y* is the reference value, *ŷ* is the measured value, *ȳ* is the mean of the reference value, and *N* is the number of measurements.

## 3. Results and Discussion

### 3.1. ATMOS41 Inter-Sensor Variability

Instrument orientation data were recorded in the X- and Y-orientation for all three ATMOS41 stations to identify undesired rotation or tilt. Orientation data (Figure 2) showed that all stations remained stable within ±2 degrees of dead level in X- and Y-direction as recommended for accurate measurements in the user manual [41]. A few larger tilts that exceed the ±2 degrees mark are observed in Figure 2, which mostly coincide with wind speeds >6 m/s (data not shown). However, only ~0.3% of measurements were affected for Atmos2 and Atmos3 and large tilts were never sustained for more than a few measured time steps. For Atmos1, a larger 2.6% of measurements were affected due to a small, temporary change in orientation between 24 and 29 April 2020, which was likely caused by a movement of the whole mounting structure. In addition, Atmos1 showed a slight misalignment of 0.5 to 1.0 degrees compared to Atmos2 and Atmos3, which was not considered significant.

The inter-sensor variability of the three ATMOS41 stations was analysed for the entire observation period (23 April to 5 July 2020) for all standard weather variables by examining 10 min instantaneous data. Figure 3 shows a pairwise comparison of the three stations using scatterplots, histograms with probability density functions, and the R^2^ value arranged in a matrix. The scatterplots show good agreement and no apparent bias between stations, with most of the data points lying in the proximity of the identity line. Some scattering effect can be observed for solar radiation (Figure 3a), which may have been caused by temporal shading of a single sensor or differences in response time to changing radiation. Relatively strong scatter can be observed in the wind speed measurements (Figure 3f), which was likely caused by other external effects such as small-scale turbulences around the stations. This scatter is reduced considerably when the data are aggregated to a larger time step (data not shown). The histograms and probability density functions of all measured variables generally show very similar distributions. Only in the case of relative humidity (Figure 3e) does Atmos1 show small differences in the distribution of values compared to the histograms of Atmos2 and Atmos3.

The comparison of all variables shows an R^2^ ≥ 0.96 except for wind speed, for which the R^2^ ranges between 0.72 and 0.74. R^2^ values increase when hourly averages are considered (data not shown), especially in the case of wind speed (R^2^ increases to 0.91 for Atmos1 vs. Atmos2, 0.92 for Atmos1 vs. Atmos3, and 0.90 for Atmos2 vs. Atmos3). Despite most comparisons being rather satisfactory, there is slightly better agreement between Atmos2 and Atmos3 when compared to Atmos1 for solar radiation, atmospheric pressure, and relative humidity.

A statistical summary with a pairwise assessment of all three ATMOS41 stations is given in Table 3. There is generally close agreement between all stations for most parameters with low RMSE and small MBE. Larger variability within the three stations was observed for wind speed and precipitation measurements. RMSE for wind speed is ~0.76 m/s at an average wind speed between 2.02 and 2.11 m/s. Atmos1 and Atmos2 measured on average slightly higher wind speed compared to Atmos3 as shown by the mean and MBE. Precipitation measurements show a RMSE of ~0.06 mm at an average precipitation between 0.17 and 0.20 mm. The variability in precipitation measurements becomes more apparent when comparing the total precipitation amounts, which were unusually low for the observed months from late April to early July. The total amounts are 82.21 mm (Atmos1), 75.92 mm (Atmos2), and 70.79 mm (Atmos3), while long-term monthly means (1981–2010) are between 47 and 77 mm for the same months [44]. The difference between the three stations is considerable given the relatively short observation period and low total rainfall and stands in contrast to the test measurements performed by METER, where a difference of <20 mm was observed within three ATMOS41 stations for a total of ~800 mm of rainfall over a period of 4 months [32]. The results suggest that wind-induced random errors such as the deflection of air flow and the formation of eddies and turbulences around the gauges [45] had an important effect on the measurements. Atmos1 was positioned west-southwest of the other two stations, which was identified as the prominent wind direction during rainfall (data not shown). The three stations may have perturbed each other due to their alignment with respect to the wind direction and the relatively small distance between the stations, thus increasing the above-mentioned wind effects for Atmos2 and even more for Atmos3. This could explain the consistently lower amounts of rainfall measured by Atmos2 and Atmos3 compared to Atmos1. Low rainfall rates, as observed for most of the measurement period, show a high volumetric fraction of smaller drops (diameter < 1 mm), which are particularly prone to wind induced errors [46]. This may have caused the large observed variability despite the relatively low wind speeds observed during rainfall events and throughout the measurement period (~2 m/s).

Generally, somewhat lower RMSE and MBE were observed between Atmos2 and Atmos3 as opposed to Atmos1 for solar radiation, atmospheric pressure, and relative humidity. The greater similarity between the newer ATMOS41 variants with regard to the latter two variables is most likely a result of the sensor improvements implemented after 2017, as mentioned above. However, the most pronounced difference was observed for solar radiation, where a bias of ~−25 W/m^2^ was found between the older Atmos1 (2017 version) and the newer ATMOS41 stations. In comparison, the bias between Atmos2 and Atmos3 was only −0.39 W/m^2^ (Table 3).

At first, the ageing of the pyranometer was considered as a possible explanation for the better agreement between the two newer ATMOS41 stations. This assumption was tested using previous data from the older Atmos1 (2017 version). Between 12 December 2017 and 24 May 2018 (164 days), the station was set up next to the ICOS site in Selhausen, 350 m from the ICOS-bkp station (Figure 1). Graphical and statistical analysis showed minor differences in the performance of the station between the two periods (data not shown), which is more likely a result of the different seasons and lengths of the two observation periods. The results suggest a stable performance of the Atmos1 over the 3-year period, even though calibration or maintenance were not performed. However, Atmos1 did not operate continuously throughout this period and hence it was not exposed to adverse weather conditions, such as strong solar radiation or heavy wind and precipitation. Therefore, sensor ageing or deterioration should be further studied, especially when continuous deployment of the station as part of a large monitoring network such as TAHMO is intended. A long-term assessment could include field visits, calibration checks and the establishment of statistical validation procedures as proposed in [47] or, if possible, comparison with a nearby reference station over an extended period.

Communication with the manufacturer allowed us to identify another possible issue related to the pyranometer provided by Apogee Instruments. A problem in the production of the early pyranometers was identified, which affected some of the earlier weather stations and was solved at a later stage. This most likely explains the observed difference in performance between the older Atmos1 (2017 version) and the more recent Atmos2 and Atmos3 stations.

### 3.2. Comparison of ATMOS41 with ICOS Backup Station

In the following, data collected over the 73-day period that includes late spring and early summer months with a small data gap of two days in mid-June are compared. The first three weeks of radiation data for Atmos3 were missing due to a defect funnel that was later replaced. To better visualize the comparison of the different stations, only a period of eight days from 30 May to 6 June (23 April to 5 July for precipitation) is shown in this section. The full time series can be found in the appendix (Appendix A).

Table 4 shows a summary of the statistical performance analysis of the three ATMOS41 stations compared to the reference station. Overall, R^2^ > 0.90 and relatively low RMSE, MBE and MAE were found for most variables except precipitation, wind speed, atmospheric pressure and solar radiation (only Atmos1). In the following, each variable is assessed in more detail.

#### 3.2.1. Solar Radiation

Figure 4a shows an 8-day period of solar radiation as measured by the four weather stations. The timing and variability of radiation during the day are well captured by the ATMOS41 stations. However, the maximum measured solar radiation is slightly lower than that of the reference station, especially for Atmos1. On a clear day, Atmos3 shows a recurring small drop in solar radiation in the early morning, suggesting a shadow cast from a surrounding sensor. On overcast days such as 4 June, the four stations show almost identical measurements.

The scatterplots of the ATMOS41 station vs. the ICOS-bkp (Figure 4b–d) confirm the overall good agreement of the stations, with an R^2^ between 0.96 and 0.99 (Table 4). The plots show little scatter and RMSE is ~32 W/m^2^ for Atmos2 and Atmos3 and somewhat higher for Atmos1 (56.46 W/m^2^) (Table 4). Solar radiation values >400 W/m^2^ show a small underestimation by the ATMOS41 (Figure 4b–d).

Figure 4e depicts the deviation between the three ATMOS41 stations and the ICOS-bkp station through a probability density plot of the residuals from hourly average data, which considers only daytime solar radiation. The peaks of the distributions show a small tendency of the ATMOS41 stations to measure higher values (negative residuals), which occurs at lower solar radiation as suggested by the scatterplots (Figure 4b–d). The underestimation of high solar radiation is represented in the right tail of the distribution (positive residuals), with a mean bias of −35.22 W/m^2^ for Atmos1 and mean biases of −9.03 and −10.06 W/m^2^ for Atmos2 and Atmos3, respectively (Table 4).

The presented results for the Atmos1 generally agree well with the analysis by [33], which compared the 2017 version of the ATMOS41 station with a SwissMetNet station. In their study, a lower bias of 8.9% was found compared to the one in this comparison (9.9%). This may be attributed to the overall lower radiation during the winter period studied by [33] as opposed to the early summer period of this study that included many sunny days. Despite the 2 km distance between pyranometers, the authors observed a lower MAE and RMSE (13.57 and 39.40 W/m^2^) than what was found in this study, which may again be related to the characteristics of the observation period since the ATMOS41 measures more accurately in the lower radiation range.

Despite the small systematic deviation from the reference ICOS-bkp station, the quality of the radiation measurements provided by the ATMOS41 was satisfactory. The newer stations show considerable improvement compared to the 2017 version of the station (Atmos1) and confirm the comparison test performed by the manufacturer, where a linear regression (y = 1.0323x) showed ~3% underestimation [32]. Linear regressions for Atmos2 (y = 1.0372x) and Atmos3 (y = 1.0336) were similar to the one found by METER (Appendix A). Granting that this bias persists in other climates and locations and compared to other high-performance pyranometers, a simple linear correction function may be developed and used to adjust the measurements.

#### 3.2.2. Precipitation

Figure 5a shows an 8-day period with several precipitation events between 28 April and 4 May. The timing of the events agrees well for all four stations, but there are some differences in magnitude and the effect of the different measurement resolutions (0.017 mm for the ATMOS41 and 0.05 mm within an hour for the Pluvio^2^ that is used at the ICOS-bkp station) is visible. A direct comparison of the rainfall measured by the two gauges is complicated given the difference in measurement resolution, gauge size and shape, and installation height, as well as the use of a windshield with the Pluvio^2^. The difference in resolution caused a greater scatter for small rainfall amounts in the 10 min time series (Figure 5b), with an R^2^ ~0.9, RMSE ~0.15, and MAE ~0.10 mm for the three stations. The event-based analysis compared 46 rainfall events with rainfall amounts ranging between 0.2 and 19.5 mm and showed more coherent results with R^2^ ~0.99 (Figure 5c). On average, Atmos1 measured higher precipitation, Atmos3 measured somewhat lower precipitation, while Atmos2 showed the least bias compared to the reference station (Table 4).

Differences between the stations are more apparent when the cumulative precipitation for the observation period is analysed (Figure 5d). Total differences in precipitation compared to the reference are 5.78 mm (7.56%), −0.51 mm (−0.67%), and −5.64 mm (−7.38%) for Atmos1, Atmos2, and Atmos3. The difference to the reference rain gauge and between the ATMOS41 stations (as discussed in Section 3.1) is considerable and shows higher discrepancies than what is reported by the manufacturer (within 3% of the average of three tipping-spoon rain gauges) [32]. Surprisingly, [33] found an underestimation of only 8.7%, even though their observation period included the entire winter season with several snowfall events. Since the ATMOS41 rain gauge is not heated and solid precipitation first needs to melt before it can be measured, higher errors could be expected during that period. This could not be further investigated, since snow was not observed during the measurement period of the present study. However, many applications such as agricultural monitoring or the use of the station in snow free climates do not rely on accurate measurements of the volume of solid precipitation.

As previously discussed in Section 3.1, wind-induced errors have likely played an important role in the measurement of rainfall, leading to significant errors considering the relatively small total precipitation amount and low rainfall intensities that were characteristic for the observed period. Additionally, gauge size and shape influence the deformation of the wind field at the gauge and minor changes in installation height can cause differences of up to 10% in precipitation measurements, as comparison studies of different rainfall gauges have shown [46,48]. A higher wind-induced under catch could therefore be expected for the ATMOS41 stations that were installed at an approximate height of 2 m compared to the Pluvio^2^ that is installed at a height of 1 m and uses an Alter windshield which has shown to improve the performance of the gauge [49,50]. The higher precipitation amount measured by the Atmos1 could be a result of the frequent detection of very small rainfall amounts, since the Pluvio^2^ does not measure fine precipitation below a threshold of 0.05 mm within an hour.

Rainfall intensity during the observation period rarely exceeded 10 mm/h, a commonly used threshold for heavy rainfall [51]. Those events did not show lower accuracy of the ATMOS41 station, but a longer observation period with higher rainfall intensities is needed to accurately assess the performance of the station during extreme events.

#### 3.2.3. Air Temperature

Figure 6a shows air temperature data of the four stations during an 8-day period. Temperature dynamics are well captured by all ATMOS41 stations. However, daily maximum temperature and temperature during rainfall (5 June) are slightly lower and show a higher noise level for the ATMOS41 stations. The latter could be a result of a wet, exposed temperature sensor or its immediate surroundings, making it more prone to evaporative cooling compared to the shielded ICOS-bkp sensor. In comparison, [33] found that night-time lows measured by the ATMOS41 were generally lower compared to the IAC instrument, while showing high relative humidity. The authors observed temperatures ranging from −13 to 23 °C with a mean temperature of 4.5 °C, as opposed to the mean temperature of 15 °C measured during the present study. The scatterplots (Figure 6b–d) and statistical analysis (Table 4) show very good performance of the ATMOS41 with values close to the identity line, little scatter, and R^2^ close to 1. RMSE and MAE are between 0.33 and 0.53 °C for all stations, nearly 50% lower than the RMSE and MAE reported in [33].

Similar to the findings of [33], there is a small mean bias towards lower temperature measured by the ATMOS41 (MBE between −0.16 and −0.37 °C), as also reflected in the probability density plot of the hourly residuals (Figure 6e). The temperature sensor of the ATMOS41 is exposed to solar heating, which is why an energy balance correction is used to calculate the actual temperature. The correction factor is proportional to solar radiation and inversely proportional to wind speed. Since errors in the measurement of those two variables may propagate to the temperature measurement, the overestimation of wind speed may explain the small bias in the measurement (Table 4). However, most values lie within 0.5 °C difference. Additionally, no tendency to lower accuracy with temperatures >30 °C was identified, which suggests that the ATMOS41 measurements are reliable within the observed range of −1.1 to 32.2 °C. Even though the accuracy of ±0.6 °C, as stated by the manufacturer, does not meet the “achievable uncertainty” standard of ±0.2 °C used by ICOS, air temperature measurements with the ATMOS41 were reliable and consistent.

#### 3.2.4. Atmospheric Pressure

Figure 7a shows atmospheric pressure measured by the four stations during an 8-day period. The ATMOS41 stations closely follow the reference station with small differences that are consistently found during daily peaks and at lower pressures, which generally coincide with rainfall. The high R^2^ ≥ 0.97 indicates good agreement of the measurements. However, RMSE and MAE are relatively large, ranging between 0.75 and 1.17 hPa and 0.64 and 1.02 hPa, respectively. In agreement with [33], the scatterplots (Figure 7b–d) and the probability density plot (Figure 7e) show a small bias towards higher values measured by the ATMOS41 compared to the reference station (MBE between 0.63 and 1.01 hPa). Atmos1 shows slightly lower overall performance, which was likely improved as a consequence of the secondary calibration added for the newer stations (see Section 1). While the ATMOS41 performs satisfactorily within the manufacturer stated accuracy of ±1 hPa, the pressure sensor does not meet the “achievable uncertainty” requirement of 0.3 hPa as commissioned by the WMO [37]. Therefore, the ATMOS41 shows only moderate performance in measuring atmospheric pressure compared to the reference station.

#### 3.2.5. Relative Humidity

Figure 8a shows relative humidity as measured by all four stations during an 8-day period of the measured time series. Relative humidity is captured well by the ATMOS41, with slightly higher humidity measured only during rain events such as 5 June for all ATMOS41 stations. This matches the observed small underestimation of temperature during rain events, as discussed in Section 3.2.3. Atmos1 additionally shows higher values during the daytime minimum humidity. The statistical summary (Table 4) shows R^2^ ≥ 0.95 for all stations and RMSE and MAE range from 3.4 to 4.3% and 2.5 to 3.5%, respectively, with Atmos1 showing slightly poorer performance than Atmos2 and Atmos3.

The scatterplot for Atmos1 (Figure 8b) confirms a small bias towards higher values for lower relative humidity and towards lower values when humidity is high. As a result, the Atmos1 shows a relatively higher MBE of 1.37% compared to Atmos2 and Atmos3 (MBE of 0.25 and −0.36%, respectively). This indicates that the manufacturer’s adaptation of the calibration function (see Section 3.1) for the newer stations resulted in an improvement compared to the older Atmos1 (2017 version). The probability density plot of the residuals (Figure 8e) confirms the improved performance of the newer stations.

The ATMOS41 stations tend to saturate at 100% relative humidity more frequently than the reference station, which seems to verify the observation of [33] and which may also be related to the underestimation of air temperature, as discussed in Section 3.2.3.

#### 3.2.6. Wind Speed and Direction

Figure 9a shows wind speed measured by the four stations during an 8-day period. Daily wind dynamics measured by the three ATMOS41 stations match well with the measurements of the Vaisala station. However, measurements by the ATMOS41 show higher peak values and a larger variability compared to the Vaisala station, which can be explained by the finer resolution of the anemometer of the ATMOS41.

The scatterplots (Figure 9b–d) show relatively large scatter around the identity line, with an R^2^ between 0.58 and 0.63. The wide spread in wind measurements is likely a result of small-scale turbulence caused by surrounding instruments, as discussed in the context of the precipitation measurements in Section 3.2.2 and which are captured due to the rapid response of ultrasonic anemometers to sudden changes in wind speed [52]. R^2^ increases up to 0.89 when hourly averages are considered, suggesting that the scatter can be reduced when small-scale differences average out over larger periods. RMSE and MAE are ~0.9 and ~0.6 m/s, respectively (Table 4). The probability density plot of the residuals (Figure 9e) shows a small mean overestimation of wind speed (negative residuals) with MBE between 0.09 and 0.18 m/s, with Atmos3 showing the best performance. Both station types used in this comparison use ultrasonic anemometers, which can measure very low wind speeds. Therefore, the agreement found in this comparison was higher than that of [33], where the ATMOS41 was compared to a cup anemometer that records zero wind speed values more frequently.

Wind direction was compared by drawing wind roses for each station (Figure 10a–d), where the length of the bins represents the frequency of the observed direction in percent, while colours indicate the magnitude of wind speed. West to South-West and East are dominant wind directions that occur, in total, ~40% of the time with a top frequency of around 7.5% for West/South-West, while wind from the North is observed in total ~13% of the time. The measurements from the Vaisala station agree well with the commonly observed wind direction at the Selhausen site [35]. Strong winds were mainly observed from West and South-West and sometimes from the North, while East winds were considerably weaker. Wind roses from the ATMOS41 stations agree in the main wind directions and speed with the reference station. Atmos1 more frequently recorded northerly winds with a top frequency of ~7%, while Atmos2 and Atmos3 recorded West/South-West winds with a higher frequency of ~10% as compared to the reference station. Wind roses for the newer ATMOS41 stations differ somewhat from that of Atmos1 likely due to adjustments made by the manufacturer (Section 3.1). Although our results do not show a significant improvement of the measurement from the older Atmos1 (2017 version) to the newer ATMOS41 stations, wind direction is still measured reasonably well by the ATMOS41.

## 4. Conclusions

This study evaluated the performance of the ATMOS41 all-in-one weather station over a period of 73 days by assessing the inter-sensor variability of three stations and by comparison against high quality, highly standardized reference meteorological stations. Inter-sensor comparison of the three ATMOS41 stations showed overall close agreement for most variables, while the newer Atmos2 and Atmos3 stations performed better in measuring atmospheric pressure, relative humidity and solar radiation compared to the older Atmos1 (2017 version). Solar radiation showed the greatest improvement, where the bias was reduced from 35.22 W/m^2^ to ~9.55 W/m^2^. Generally good agreement with R^2^ > 0.95 and small biases were observed for most of the examined weather variables when compared to the reference station. If reference solar radiation data are locally available, a simple linear correction function was proposed to account for the 3% systematic bias that remained in solar radiation measured by the ATMOS41. The atmospheric pressure sensor of the ATMOS41 showed only moderate performance compared to the ICOS station, showing greater uncertainty in the measurements than recommended by the “achievable uncertainty” standard commissioned by the WMO. The measurement of wind speed by the ATMOS41 was slightly overestimated and showed relatively large scatter. Better results are achieved with hourly or half-hourly averages, which are suitable for most modelling applications. The largest variability between the stations was found in the measurement of precipitation, where total precipitation measured by the ATMOS41 showed differences around ±7.5% compared to the reference. This was attributed mainly to wind-induced errors that may have been exacerbated due to the close proximity of the three ATMOS41 stations as well as differences in the measurement resolution and architecture of the compared rain gauges.

The results of this study showed similar or improved performance of the ATMOS41 compared to the early performance test, but also revealed its limitations. Further work should focus on the performance assessment of the ATMOS41 during extreme precipitation and wind speed as well as the long-term durability and accuracy of the station. The station seems to be well suited for private users. In particular, farmers in high-income countries can benefit from its compact design and limited maintenance requirements. Developing countries may similarly benefit from the ATMOS41 station when costs are jointly carried by multiple actors that use the collected data to market data products to private and governmental institutions. This strategy is applied within the TAHMO project. Due to the higher uncertainty related to atmospheric pressure and precipitation measurements, and the non-heated gauge, the use of the ATMOS41 station in research appears to be better suited for studies where the amount of solid precipitation is not relevant, where precise rainfall or atmospheric pressure is not a key parameter or when multiple gauges can be deployed to calculate average values for a given location. Overall, the ATMOS41 is a good compromise between measurement accuracy and cost effectiveness, making it an attractive component of wireless sensor networks as well as an expansion tool for weather monitoring networks in remote areas or under limited financial resources.

## Figures and Tables

**Figure 1 sensors-21-00741-f001:**
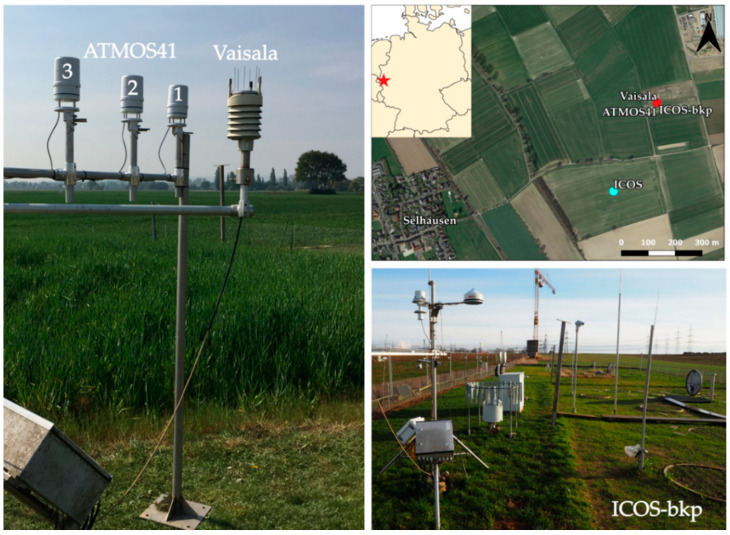
Experimental site with the three ATMOS41 stations, the Vaisala weather transmitter and the ICOS-bkp station.

**Figure 2 sensors-21-00741-f002:**
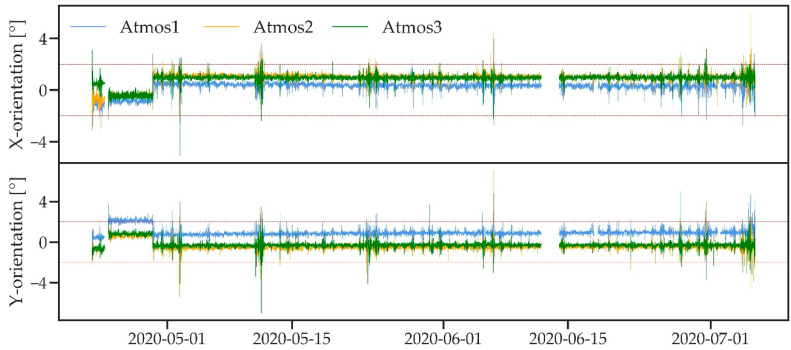
*X*- and *Y*-orientation for the three ATMOS41 weather stations. The red dotted line indicates ±2 degrees from dead level.

**Figure 3 sensors-21-00741-f003:**
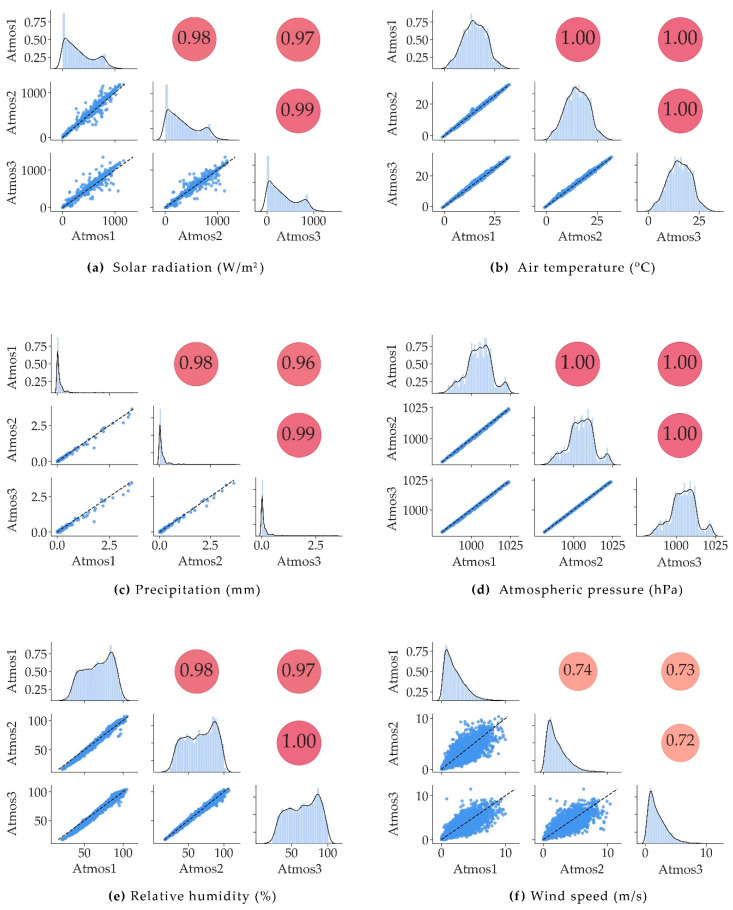
(**a**–**f**) Correlation matrices for all weather variables measured by the three ATMOS41 stations. Subplots in the lower left show scatterplots of station pairs with the dashed line indicating the 1:1 identity line, the diagonal shows histograms of measured values with probability density functions, upper right shows the coefficient of determination R^2^.

**Figure 4 sensors-21-00741-f004:**
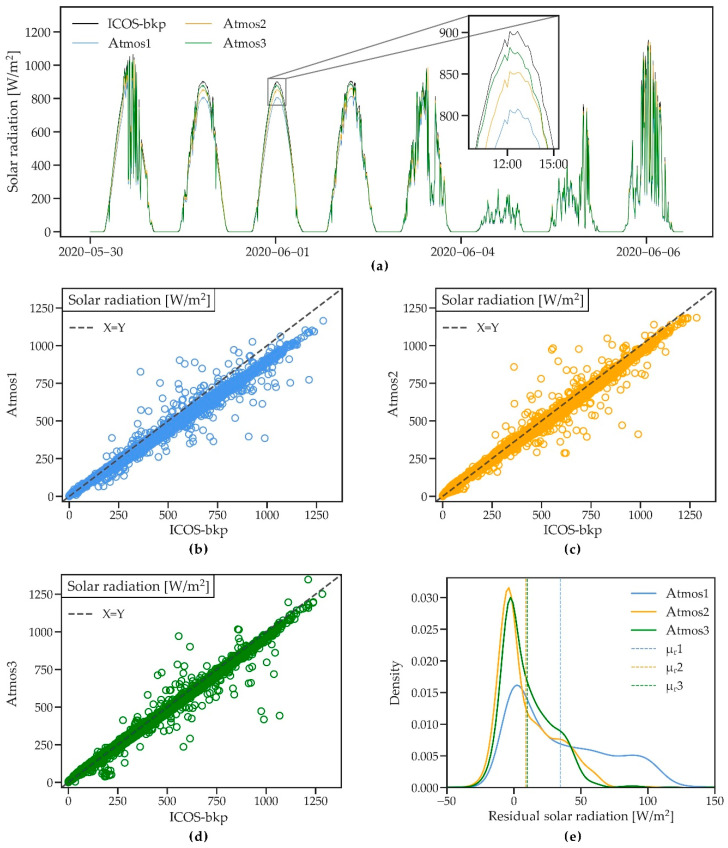
(**a**) A short time series of solar radiation measured by three ATMOS41 weather stations and the ICOS-bkp station from 30 May to 6 June 2020. (**b**–**d**) Scatterplots of 10 min solar radiation for the three ATMOS41 stations vs. the reference station. (**e**) Probability density functions of the residual mean hourly solar radiation. Dashed lines show the mean of residuals (μ_r_).

**Figure 5 sensors-21-00741-f005:**
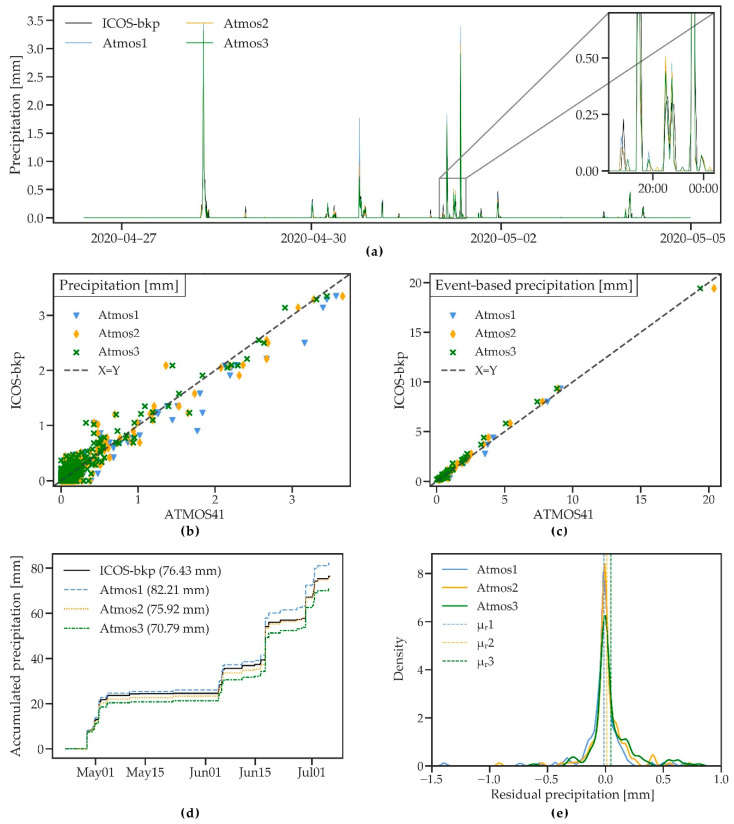
(**a**) A short time series of precipitation measured by three ATMOS41 weather stations and the ICOS-bkp station from 28 April to 4 May 2020. (**b**,**c**) Scatterplots of 10 min precipitation and event-based precipitation sum for the three ATMOS41 stations vs. the reference station. (**d**) Cumulative precipitation measured by the three ATMOS41 weather stations and the reference station for the whole time series (numbers in parentheses refer to total precipitation amount). (**e**) Probability density functions of the residual hourly precipitation sum. Dashed lines show the mean of residuals (μ_r_).

**Figure 6 sensors-21-00741-f006:**
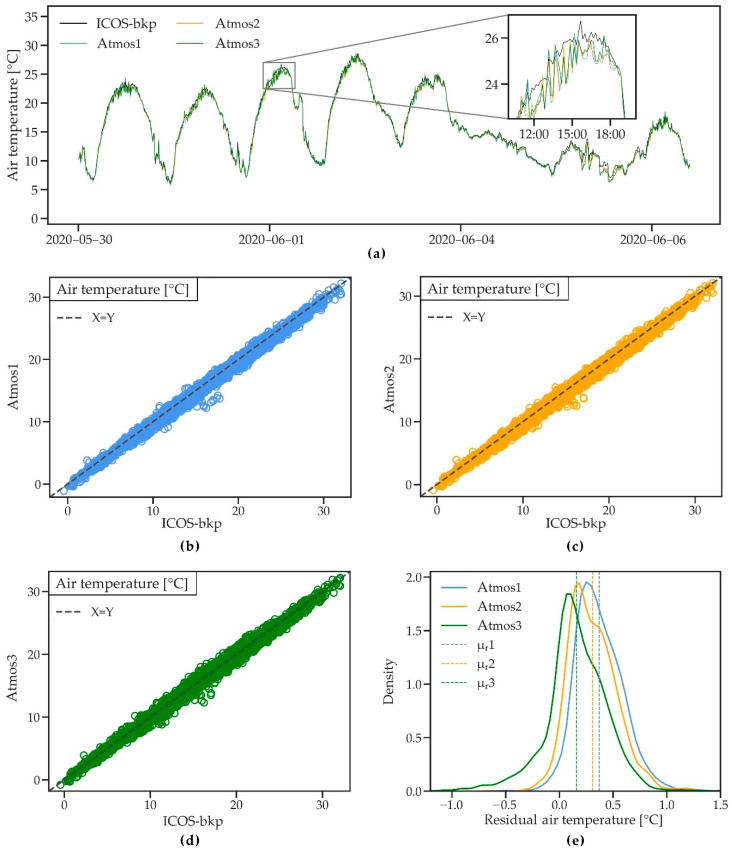
(**a**) A short time series of air temperature measured by three ATMOS41 weather stations and the ICOS-bkp station from 30 May to 6 June 2020. (**b**–**d**) Scatterplots of 10 min air temperature for the three ATMOS41 stations vs. the reference station. (**e**) Probability density functions of the residual mean hourly air temperature. Dashed lines show the mean of residuals (μ_r_).

**Figure 7 sensors-21-00741-f007:**
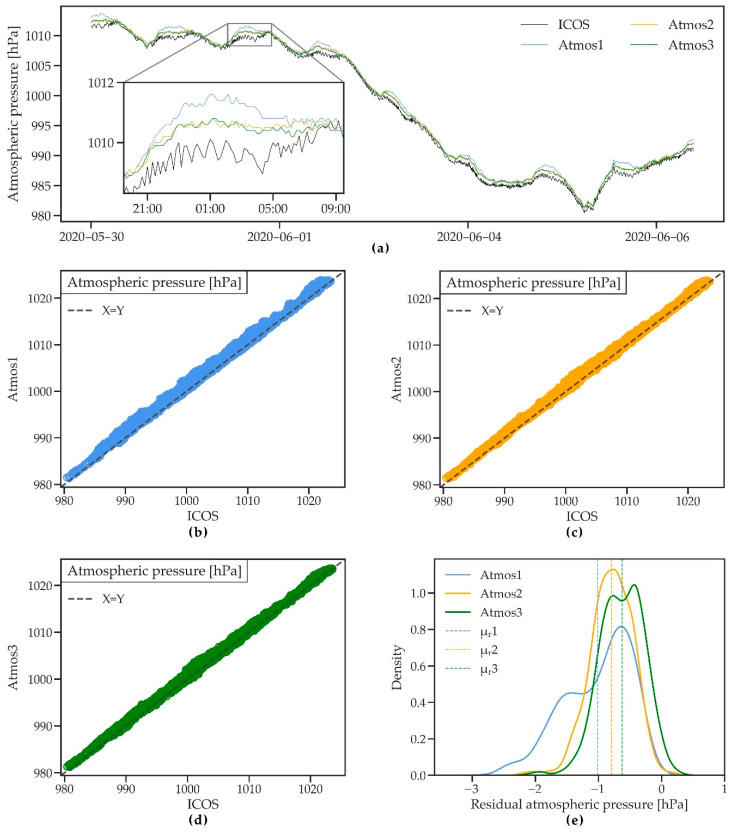
(**a**) A short time series of atmospheric pressure measured by three ATMOS41 weather stations and the ICOS-bkp station from 30 May to 6 June 2020. (**b**–**d**) Scatterplots of 10 min atmospheric pressure for the three ATMOS41 stations vs. the reference station. (**e**) Probability density functions of the residual mean hourly atmospheric pressure. Dashed lines show the mean of residuals (μ_r_).

**Figure 8 sensors-21-00741-f008:**
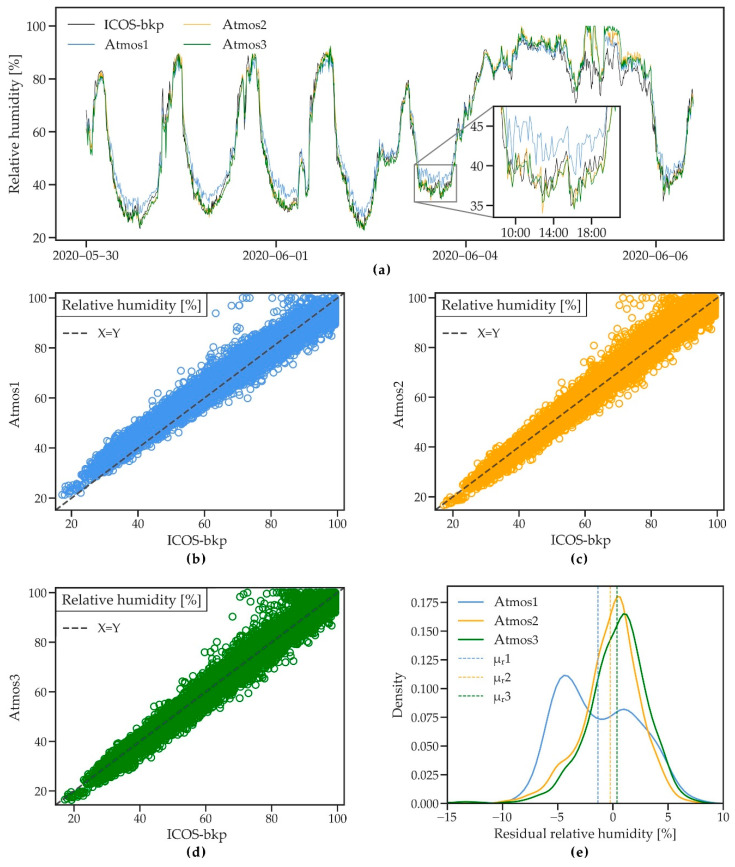
(**a**) A short time series of relative humidity measured by three ATMOS41 weather stations and the ICOS-bkp station from 30 May to 6 June 2020. (**b**–**d**) Scatterplots of 10 min relative humidity for the three ATMOS41 stations vs. the reference station. (**e**) Probability density functions of the residual mean hourly relative humidity. Dashed lines show the mean of residuals (μ_r_).

**Figure 9 sensors-21-00741-f009:**
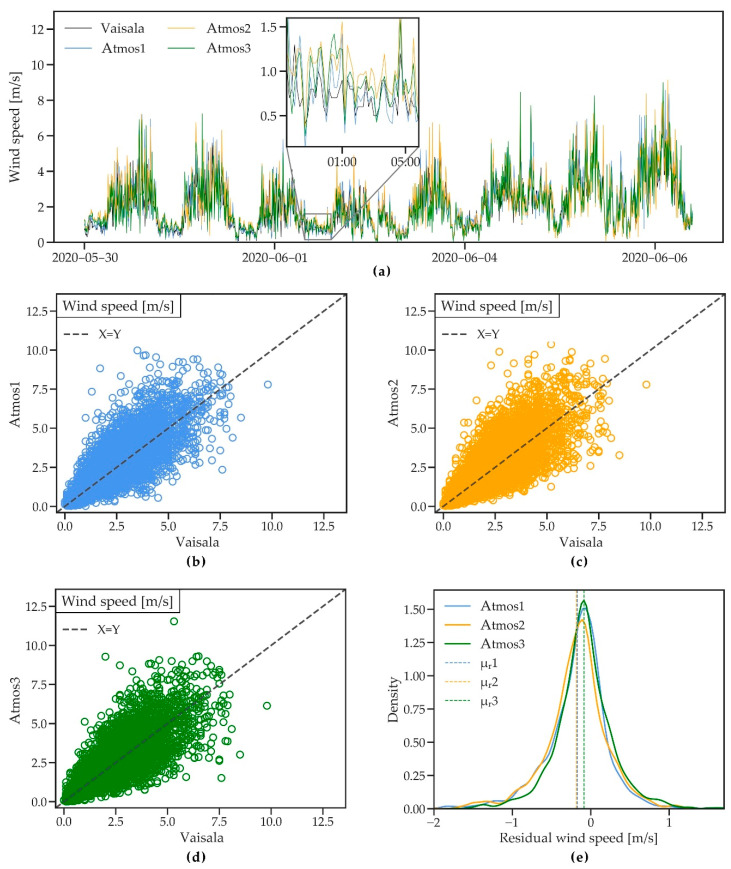
(**a**) A short time series of wind speed measured by three ATMOS41 weather stations and the ICOS-bkp station from 30 May to 6 June 2020. (**b**–**d**) Scatterplots of 10 min wind speed for the three ATMOS41 stations vs. the reference station. (**e**) Probability density functions of the residual mean hourly wind speed. Dashed lines show the mean of residuals (μ_r_).

**Figure 10 sensors-21-00741-f010:**
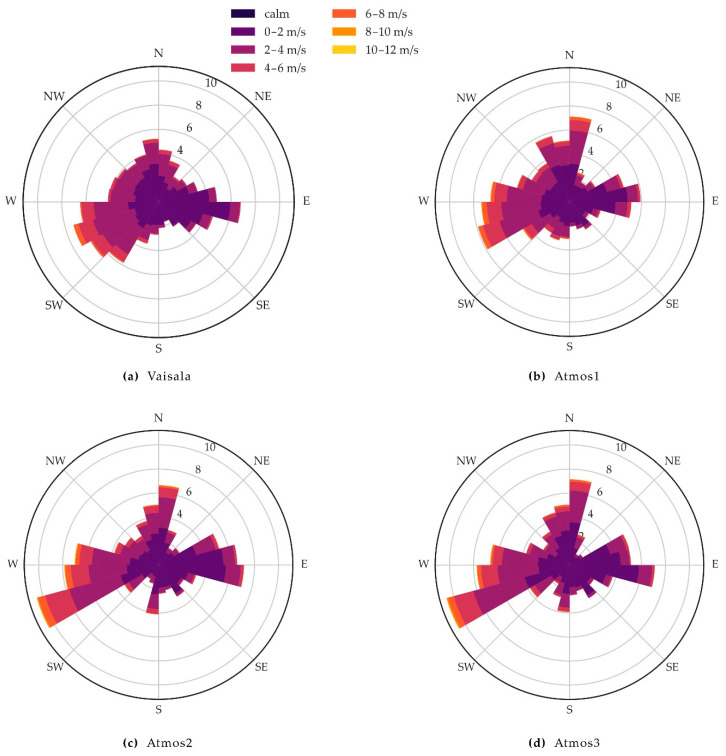
Wind roses showing the frequency of observed wind direction at a 10 min interval measured by (**a**) the Vaisala reference station. (**b**–**d**) the three ATMOS41 weather stations, for the period from 23 April to 5 July 2020.

**Table 1 sensors-21-00741-t001:** Characteristics of the ATMOS41 all-in-one weather station.

Characteristic	ATMOS41
Manufacturer	METER Group, Inc.
Cost	EUR 1750
Dimensions	Height: 34 cm, Ø = 10 cm
Warranty	1 year
Installation	Mount on pole, stand, or tripod; orient to true North; level the weather station
Maintenance	Recalibration: every 2 years; Cleaning: check for bird droppings and insect debris
Power requirements	Supply Voltage: 3.6 to 15 V Current draw: 8.0 mA during measurement, 0.3 mA while asleep
Operating temperature	−40 to +50 °C
Communication protocol	SDI-12
Additional equipment	Pole, stand or tripod and a data logger (third party loggers are compatible too)

**Table 2 sensors-21-00741-t002:** Sensor details for the ATMOS41 weather station as well as the ICOS-bkp, Integrated Carbon Observation System (ICOS) or Vaisala station.

Parameter	ATMOS41	ICOS-bkp, ICOS or Vaisala
Radiation	Miniature pyranometer with silicon-cell (Apogee Instruments, Logan, USA) Resolution: 1 W/m^2^ Accuracy: ±5%	Pyranometer with permanent ventilation/heating (CMP21, Kipp & Zonen, Delft, Netherlands; EUR 900) Resolution: 1 W/m^2^ Accuracy: ±1%
Precipitation	Optical sensor rain gauge with 68 cm^2^ catch area (METER Group Inc., Pullman, USA) Resolution: 0.017 mm Accuracy: ±5% (up to 50 mm/h)	Weighing rain gauge with 200 cm^2^ catch area (Pluvio^2^, Ott HydroMet, Kempten, Germany; EUR 5000) Resolution: 0.05 mm within an hour Accuracy: ±1 mm
Temperature	Thermistor, non-aspirated (METER Group Inc., Pullman, USA) Resolution: 0.1 °C Accuracy: ±0.6 °C	Resistance thermometer PT100 1/3 Class B (HC2S3, Rotronic, Bassersdorf, Germany; EUR 900) Resolution: 0.01 °C Accuracy: ±0.1 °C
Relative humidity	(METER Group Inc., Pullman, USA) Resolution: 0.1% Accuracy: ±3% (varies with temperature and humidity)	ROTRONIC^®^ Hygromer IN-1 (HC2S3, Rotronic, Bassersdorf, Germany) Resolution: 0.02% Accuracy: ±0.8%
Pressure	Barometric pressure sensor (METER Group Inc., Pullman, USA) Resolution: 0.1 hPa Accuracy: ±1.0 hPa	BAROCAP^®^ sensor (PTB110, Vaisala Inc., Helsinki, Finland; EUR 730) Resolution: 0.1 hPa Accuracy: ±0.3 hPa (at +20 °C)
Wind speed	Ultrasonic anemometer (METER Group Inc., Pullman, USA) Resolution: 0.01 m/s Accuracy: the greater of 0.3 m/s or 3%	WINDCAP^®^ ultrasonic transducer (WXT520, Vaisala Inc., Helsinki, Finland; EUR 2350) Resolution: 0.1 m/s Accuracy: ±3% at 10 m/s
Wind direction	Ultrasonic anemometer (METER Group Inc., Pullman, USA) Resolution: 1° Accuracy: ±5°	WINDCAP^®^ ultrasonic transducer (WXT520, Vaisala Inc., Helsinki, Finland) Resolution: 1° Accuracy: ±3°

**Table 3 sensors-21-00741-t003:** Statistical summary of the inter-sensor comparison for all standard weather variables measured by three ATMOS41 weather stations. Colours give an evaluation of the comparison, with red indicating the lowest and green the highest performance.

		μ			RMSE			MBE		
	Station °N	1	2	3	1 vs. 2	1 vs. 3	2 vs. 3	1 vs. 2	1 vs. 3	2 vs. 3
Parameter										
Solar radiation [W/m^2^]		320.26	345.21	345.6	38.34	47.37	32.76	−24.96	−25.35	−0.39
Precipitation [mm]		0.20 (82.21) *	0.18 (75.92) *	0.17 (70.79) *	0.06	0.08	0.05	0.015	0.027	0.012
Air temperature [°C]		15.05	15.11	15.26	0.22	0.38	0.34	−0.07	−0.22	−0.15
Atmospheric pressure [hPa]		1004.92	1004.70	1004.53	0.42	0.51	0.23	0.22	0.39	0.17
Relative Humidity [%]		67.35	66.18	65.56	3.26	3.49	1.38	1.17	1.79	0.62
Wind speed [m/s]		2.1	2.11	2.02	0.77	0.75	0.76	−0.009	0.089	0.098

* values in parentheses refer to the total precipitation amount during the observation period.

**Table 4 sensors-21-00741-t004:** Statistical summary of the performance of three ATMOS41 stations compared to the ICOS-bkp or Vaisala reference station. Colours give an evaluation of the comparison, with red indicating the lowest and green the highest performance.

		R^2^			RMSE			MBE			MAE		
	Station °N	1	2	3	1	2	3	1	2	3	1	2	3
Variable													
Solar radiation (W/m^2^)		0.96	0.99	0.99	56.46	31.88	32.28	−35.22	−9.03	−10.06	38.14	18.40	17.14
Precipitation (mm/10min)		0.92	0.93	0.93	0.13	0.13	0.13	0.02	−0.01	−0.02	0.08	0.09	0.08
Precipitation (mm/event)		0.99	0.99	0.99	0.19	0.24	0.30	0.06	−0.05	−0.17	0.11	0.15	0.21
Temperature (°C)		0.99	0.99	0.99	0.53	0.49	0.45	−0.37	−0.31	−0.16	0.42	0.38	0.33
Atmospheric Pressure (hPa)		0.98	0.99	1.00	1.17	0.89	0.75	1.01	0.79	0.63	1.02	0.80	0.64
Relative Humidity (%)		0.95	0.97	0.97	4.33	3.36	3.39	1.37	0.25	−0.36	3.47	2.50	2.55
Wind speed (m/s)		0.62	0.58	0.63	0.84	0.88	0.82	0.17	0.18	0.09	0.55	0.59	0.55

## Data Availability

Data collected from the ATMOS41 stations presented in this study are available on request from the corresponding author. Data from the Vaisala station are openly available at the TERENO data portal TEODOOR (https://teodoor.icg.kfa-juelich.de/) at [DOI]. Data from the ICOS and ICOS-bkp station will be available through the ICOS data portal (https://data.icos-cp.eu/portal/) or are available on request from Marius Schmidt (ma.schmidt@fz-juelich.de).

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
