# Peer review of "Performance of the ATMOS41 All-in-One Weather Station for Weather Monitoring"

_sensors, 2021, doi:10.3390/s21030741_

Round 1
Reviewer 1 Report
Peer review on manuscript “Performance of the ATMOS41 all-in-one weather station for weather monitoring”
This paper focuses on the evaluation of performance of ATMOS41 weather station for the short time period (little bit more than two months) in specific geographical area.
My first question is related to the motivation of this paper. Usually manufacturers (no matter what kind of equipment they produce) not only design, develop and sell equipment, but they also assess the performance of equipment including environmental conditions in which the equipment can perform their functions, accuracy etc. All of this information is included in the documentation. So I don’t understand why the authors tried to check the ATMOS41 manufacturer?
Second. The testing period is very short and covers only a couple of months. For comprehensive testing it is required to consider different seasons and ideally use several years of measurements.
From my viewpoint, the paper is not research paper but some kind of a weak technical report.
I am on the position to reject the paper but I would like to give authors a chance to strongly reconsider their manuscript.
Reviewer 2 Report
The article "Performance of the ATMOS41 all-in-one weather station for weather monitoring" submitted for review by Dombrowski et al. investigates the performance of "affordable" but less accurate ATMOS41 compared to high quality sensors of ICOS-bkp and Vaisala. The paper is mainly based on field measurement from 23 April to 5 July 2020 (73 days) in Selhausen, Germany. The authors developed a well constructed statistical evaluation of 7 record parameters comparing the accuracy claim of the manufacturer with 3 in-situ measurements. The arguments supporting the article are logically presented, using convincing Figures and Tables. The article presents interesting results for the users of ATMOS41 providing quantitative measurement of its bias and suggesting some solution to it. It is an investigation of good quality. A discussion part could be developed that could address the quality of the proposed setting (period of measurement, number of stations, localisation choice) and how that may impact the results. This could also provide ideas on the optimum use of those stations (for example for the cited TAHMO project) as well as priority improvement of this equipment to address its biggest bias (precipitation) if that possible.
Questions
Questions 1
"The atmospheric pressure sensor of the ATMOS41 does not meet WMO standards"
what about the other sensors, temp. Rainfall. Are there reaching WMO standards ?
Questions2
Is there any publications that were assessing the impact of rainfall measurement versus the catch area.
Questions 3
Would you think that the choice of the location of the study may impact the accuracy you obtained ?
That may also provide you with an argument to extend ATMOS testing when comparing the IAC/ETH and SwissMetNet testing.
Is it a new location with different weather conditions ? or different land surfaces for example.
Questions 4
Line 216 "82.21 mm (Atmos1), 75.92 mm (Atmos2), and 70.79 mm (Atmos3)."
Is any risk that the 3 sensors perturb each other ? as all 3 are located very close to each other ? or that some other measurement are differently affected by surrounding structure (i.e. The vaisala station show Figure 1)
Question 5
Would you have an idea that could explain the wide spread of wind speed cross plot Figure 9bcd ? which seem to increase with the intensity of the wind speed ?
Broader discussion
There a couple of reflection that could be gather in a discussion on the methods and potential application of those sensor
#1 Line 252 “Therefore, sensor ageing or deterioration should be further studied by investigating a period of 2-3 years of continuous deployment tofully assess the robustness of the ATMOS41 station.”
what could be the impact on the TAHMO dataset if stations are probably taken from the same year of construction.
Is there any possibility of comparing it to other studies if knowing the age of the station. The station makers have insight into the lifetime expected of the station.
#2 Would you consider the testing setting as optimal or it could be improved in any sense ? Testing winter period ?
#3 You may try to develop that quote in your abstract " most scientific research and consumer applications."
Would you suggest that based on your study, ATMOS41 is accurate enough for private farmers ? Inaccurate for research ? Still good potential for a developing country or still too expensive ?
#4 Would you suggest what should be improve on ATMOS41 to increase their quality without make them too expensive
Tables
Table 2
Similarly to Table 1, would you be able to give an estimation of the cost of each instrument.
Then we can compare the cost of the ATMOS41. Or simply remind in the table the 10000euros cost of the ICOS-bkp station.
Figures
Figure 2
Precise what color corresponds to which atmos. It may have dropped when adding the Figure as the label of the legend is present on the top of the Figure.
Figure 5
Try to have the legend of the Figure on the same page of the Figure like done on Figure 8.
Figure 5e
Precise maybe that a residual precipitation (may be confusing due to the negative value)
Figure 6e
Idem as Figure 5e for the air temperature
Is air temperature a standard 2m temperature record ?
precise in x- and y-axis that the icos=bkp Air temperature [in C] and the atmos* Atmospheric pressure
and its units [hPa] rather than on the legend ... if possible like Figure 5 plots.
Figure 7e
Idem as Figure 5e for the atmospheric pressure
Figure7bcd
precise in x- and y-axis that the icos=bkp Atmospheric pressure and the atmos* Atmospheric pressure
and its units [hPa] rather than on the legend ... if possible like Figure 5 plots.
Figure8bcd
Same comments as Figure 7
Figure9bcd
Same comments as Figure 7
Specific comments
Line 80
Precise if you know where ATMOS41 is deployed, mainly in developing countries ? or use by the private sector in developed countries as well. As one of the main attractivity of ATMOS41 is its low price ...
Line 94 Precise that for a developed country perspective as suggested before
The device is rather inexpensive
-->
The device is rather inexpensive in developed country
Line 106
ICOS standard ? also meaning WMO standard.
Line 117 It was indicated that Atmospheric pressure, wind speed, and wind direction are by default not recorded at the ICOS-112bkpstation meaning there is no ICOS standard for that parameter. Maybe adding WMO is clearer (just a suggestion).
ICOS
-->
ICOS/WMO
Line 140
Is R was used as well ?
Line 126-136
Could you precise if the 3 ATMOS41 stations are exactly the same model and no material update was made on it. It means that difference would simply be the period of use ...
Line320
There a “2” subscripts at Pluvio2 ?
In Table 2, it is defined without subscript. Choose one to the other unless it presents different things.
You may define what Pluvio2 meaning (i.e. the rainfall sensor on ICOS-bkp station ??)
Line 333-335
"Since the ATMOS41 rain gauge is not heated and solid precipitation first needs to melt before it can be measured, higher errors could be expected during that period."
You may be precise if snow was observed during the record period.
Line 468
to the older Atmos1 (add the yyyy of commercialisation)
Round 2
Reviewer 1 Report
I accept all of your responses.